# Design of GSH-Responsive Curcumin Nanomicelles for Oesophageal Cancer Therapy

**DOI:** 10.3390/pharmaceutics14091802

**Published:** 2022-08-27

**Authors:** Zhaoming Ma, Xuzhu Gao, Faisal Raza, Hajra Zafar, Guanhong Huang, Yunyun Yang, Feng Shi, Deqiang Wang, Xia He

**Affiliations:** 1Department of Radiotherapy, The Second People’s Hospital of Lianyungang, Lianyungang 222023, China; 2Department of Radiation Oncology, Affiliated Cancer Hospital of Nanjing Medical University, Jiangsu Cancer Hospital, Jiangsu Institute of Cancer Research, Nanjing 210009, China; 3School of Pharmacy, Shanghai Jiao Tong University, Shanghai 200240, China; 4Institute of Digestive Diseases, Jiangsu University, Zhenjiang 212001, China; 5Department of Medical Oncology, Affiliated Hospital of Jiangsu University, Zhenjiang 212001, China

**Keywords:** oesophageal cancer, curcumin, nanomicelles, GSH responsive

## Abstract

Oesophageal cancer is a malignant tumor with high morbidity and mortality. Surgical treatment, radiotherapy, and chemotherapy are the most common treatment methods for oesophageal cancer. However, traditional chemotherapy drugs have poor targeting performance and cause serious adverse drug reactions. In this study, a GSH-sensitive material, ATRA-SS-HA, was developed and self-assembled with curcumin, a natural polyphenol antitumor drug, into nanomicelles Cur@ATRA-SS-HA. The micelles had a suitable particle size, excellent drug loading, encapsulation rate, stability, biocompatibility, and stable release behaviour. In the tumor microenvironment, GSH induced disulfide bond rupture in Cur@ATRA-SS-HA and promoted the release of curcumin, improving tumor targeting. Following GSH-induced release, the curcumin IC50 value was significantly lower than that of free curcumin and better than that of 5-FU. In vivo pharmacokinetic experiments showed that the drug-loaded nanomicelles exhibited better metabolic behaviour than free drugs, which greatly increased the blood concentration of curcumin and increased the half-life of the drug. The design of the nanomicelle provides a novel clinical treatment for oesophageal cancer.

## 1. Introduction

Oesophageal cancer is one of the world’s most lethal cancers. According to the latest data, the incidence of oesophageal cancer ranks 8th among all cancers in the world and its mortality ranks 6th. Furthermore, most patients diagnosed with oesophageal cancer in the clinic are diagnosed with advanced cancer. Patients with advanced oesophageal cancer have a poor prognosis and high mortality. Oesophageal cancer is mainly divided into squamous cell carcinoma and adenocarcinoma, based on their histopathology. The incidence of oesophageal cancer varies significantly based on geographical region and histopathology. Most oesophageal cancer in Western countries is oesophageal adenocarcinoma, and its incidence is increasing year by year, whereas most oesophageal cancer in China is oesophageal squamous cell carcinoma. Although the drugs for the treatment of oesophageal cancer are constantly improving, the 5-year survival rate for patients with advanced oesophageal cancer is only 5% [1,2]. Existing treatments for oesophageal cancer mainly include surgery, radiotherapy, and chemotherapy [3,4,5]. In the early stages of oesophageal cancer, the tumor appears in the axial membrane of the oesophagus, and the patients present with symptoms such as indigestion, discomfort posterior to the sternum, and poor swallowing. The most effective treatment for oesophageal cancer during this period is tumor resection, and the appropriate surgical method is selected according to the location of the tumor, the presence of lymph metastasis, and the overall status of the patient. Radiotherapy or chemotherapy is given both before and after surgery to reduce tumor volume and inhibit tumor metastasis. Adjuvant therapy improves the survival rate of patients. Commonly used oesophageal cancer chemotherapy drugs include 5-fluorouracil (5-FU) and cisplatin [6]. However, these traditional treatments have poor therapeutic effects and serious adverse effects. Patients are prone to brain metastasis following surgery, which seriously threatens their lives [7,8,9]. Therefore, safer and more efficient therapeutic drugs and improved drug targeting to the tumor are needed.

Curcumin (Cur) is a natural polyphenol extracted from the rhizome of Curcuma longa that has antioxidant, anti-inflammatory, antibacterial, antiviral, and other biological functions. It is widely used in the treatment of tumors, cardiovascular diseases, inflammation, and neurodegenerative diseases [10,11]. Evidence has shown that Cur has the potential to become an excellent antitumor drug and has a wide range of therapeutic targets, such as transcription factors STAT3, Notch1, NF-κB, growth factors TGF-β, VEFG, EGF, FGF, inflammatory factors TNF-α, COX-2, MCP-1, apoptosis-related proteins Bax, PARP, Caspase3, protein kinase MAPK, ERK, JNK, IKK, survival/proliferation-promoting proteins Bcl-2, Cyclin D1, cMyc, etc. [11]. Modulation of these targets by Curcumin may effectively inhibit tumor proliferation and induce tumor cell apoptosis and has great therapeutic potential for breast cancer [12], lung cancer [13] and gastric cancer [14]. However, the poor water solubility and low bioavailability of Cur limits its application in clinical treatment. In clinical trials, it has been reported that human oral administration of Cur at 8 g/d results in the rapid metabolism of Cur into its metabolites, resulting in a low level of free Cur in plasma (<2.5 ng/mL) [15]. Therefore, designing a drug carrier of Cur to improve its solubility and the targeting of oesophageal cancer is necessary.

Hyaluronic acid (HA) is a natural linear polysaccharide composed of the alternating disaccharide units D-glucuronic acid and N-acetylglucosamine, as proposed from bovine vitreous in 1934. In the human body, endogenous HA is an important part of the extracellular matrix and widely exists in various tissues of vertebrates. It plays an important role in cellular differentiation, adhesion, and matrix modification [16]. CD44 is a HA receptor that is overexpressed in a variety of tumor cells and acts as a characteristic tumor marker [17]. HA has good biodegradability, biocompatibility, nontoxicity, and nonimmunogenicity. Therefore, as a drug carrier of Curcumin, HA can accurately target tumors and improve the tumor uptake of Curcumin. In addition, HA has the following advantages: (1) hydrophilicity helps it form an anti-protein adsorption protective layer on the surface of the nanocarrier; (2) anionic properties (pKa = 3.4) enable it to interact with cationic polymers and liposomes to form diverse nanostructures; and (3) its reactive chemical functional groups (carboxyl and hydroxyl) provide HA derivative possibilities [18].

In this paper, Curcumin-loaded nanomicelles were designed based on HA. Since HA is a hydrophilic substance, it was necessary to modify the hydrophobic groups of HA for the compound to self-assemble into nanomicelles. Cur was encapsulated in the hydrophobic core of the micelles. All-trans retinoic acid (ATRA) is a hydrophobic oxidative metabolite of vitamin A that can induce tumor cell differentiation and inhibit tumor cell proliferation and migration [19,20]. Therefore, in the current study, ATRA was used as the hydrophobic group of nanomicelles with the hope that the combination of ATRA and Cur would improve the therapeutic effect of the nanomicelles on tumors.

After determining the main components of the nanomicelles and determining that the micelles accurately targeted tumor sites, the responsiveness of the micelles in the tumor microenvironment was improved to increase drug release into the tumor tissues. Compared with those of normal tissues, tumor metabolism and proliferation are abnormal, and tumors possess a complex and highly heterogeneous microenvironment. Tumor microenvironment characteristics mainly include the increased hypoxia, microacidity, high reduction rates, and high levels of reactive oxygen species (ROS), proteases, and esterases. Glutathione (GSH) is the most abundant thiol in mammalian cells and has strong reducibility. Glutathione/oxidized glutathione (GSH/GSSG) maintains intracellular redox balance [21]. It has been reported that the GSH concentration in the tumor microenvironment is significantly higher than that in normal tissue [22]. Under conditions of reduction, disulfide bonds are broken by high concentrations of GSH in tumor cells [23]. Thus, materials can be degraded by the breaking of disulfide bonds and chemotherapeutic drugs can be delivered into tumor cells. High concentrations of GSH in tumor cells can be rapidly depleted, thereby enhancing oxidative stress in tumor cells and sensitizing cells to drugs. In this study, cystamine-linked hydrophilic HA with disulfide bonds was combined with hydrophobic all-trans retinoic acid (ATRA-SS-HA). ATRA-SS-HA and Cur self-assembled into Cur@ATRA-SS-HA micelles that possessed tumor targeting ability. After being engulfed by tumor cells, disulfide bonds quickly responded to high concentrations of GSH in the tumor, promoting the rapid release of Cur and achieving efficient and safe tumor treatment.

## 2. Materials and Methods

### 2.1. Synthesis of ATRA-SS-HA

The main raw materials included all-trans retinoic acid, cystamine, HA (5000), EDC, triethylamine, 2-hydroxybenzotriazole (HoBt), N-hydroxysuccinimide sulfonate sodium salt Sulfo-NHS, and DMSO/H2O (1:1). In this reaction, ATRA (1.631 mmol) was dissolved in DCM, and EDC (1.957 mmol) was added. HoBt (1.957 mmol) was stirred for 2 h for activation. Cysteamine was added to another reaction flask to form a suspension. TEA (39.144 mmol) was added to the N_2_ environment and stirred for 6 h to form a white emulsion. Then, the mixture containing ATRA was slowly added to the reaction flask of cystamine for 24 h. The reaction was monitored with DCM:MeOH = 10:1 and purified by column to obtain the intermediate (114 mg). HA (0.02 mmol) was dissolved in DMSO: H_2_O = 1:1, and then EDC (0.264 mmol) and sulfo-NHS (0.264 mmol) were dissolved in the solvent and water, respectively. Then, they were added to the mixture of HA and stirred at room temperature for 2 h. The intermediate (0.264 mmol) synthesized in the previous step was added to the reaction solution, stirred in N_2_ for 24 h, dialyzed and dried, and the final product was obtained.

### 2.2. Preparation of Micelles

ATRA-SS-HA micelles loaded with Cur were prepared by dialysis. Then, 20 mg ATRA-SS-HA was dissolved in 4 mL water and stirred at room temperature for 30 min. Cur in 8 mg ethanol at a concentration of 20 mg/mL was added to the solution drop-wise, and the final solution was subjected to ultrasonic treatment for 30 min at 100 W power in an ice bath using a probe ultrasonic instrument. The solution was dialyzed with excess distilled water overnight using a dialysis bag (MWCO 3500) and then centrifuged at 3000 rpm for 10 min. The solution was filtered through a microporous membrane with a 0.45 μm pore size and freeze-dried.

### 2.3. Calculation of Critical Micelle Concentration

The critical micelle concentration of the polymer was determined by the pyrene fluorescence probe method. Pyrene (3.0 mg) was weighed precisely and transferred to a brown volumetric flask. Tetrahydrofuran was dissolved and fixed to a volume of 25 mL, and 0.12 mg/mL was obtained by shaking. The volumetric flask was moved from 1 mL to 10 mL. Tetrahydrofuran was fixed to 10 mL, and this step was repeated. Then, 1.2 × 10^−3^ mg/mL (6 × 10^−5^ mol/mL) was obtained by shaking (Pyrene tetrahydrofuran stock solution). 1 mg/mL ATRA-SS-HA and HA blank micelles were prepared via the dialysis bag hydration method. The dilutions were 1 × 10^−5^ mg/mL, 5 × 10^−5^ mg/mL, 1 × 10^−4^ mg/mL, 5 × 10^−4^ mg/mL, 1 × 10^−3^ mg/mL, 5 × 10^−3^ mg/mL, 1 × 10^−2^ mg/mL, 5 × 10^−2^ mg/mL, 1 × 10^−1^ mg/mL, 5 × 10^−1^ mg/mL, and 1 mg/mL.

A total of 40 μL of tetrahydrofuran stock solution containing pyrene was accurately added to 11 separate 10 mL EP tubes. The tetrahydrofuran was volatilized in the ventilation cabinet for 12 h to avoid light, and 4 mL of polymer micelle solution with different concentrations was added to make the final concentration of pyrene at 6 × 10^−7^ mol/mL. Ultrasonication was performed for 30 min in the dark at room temperature for 12 h. The excitation wavelength was set to 338 nm, and the excitation slit was set to 3 nm. The emission slit was set to 5 nm. The emission spectra of micelles with different concentrations were scanned by a spectrophotometer at 300–500 nm, and the fluorescence intensities at 373 nm and 383 nm were recorded. Origin software was used to draw the tangent between I373/383 and the logarithm of micelle concentration Log c to find the intersection point. The concentration at the intersection point is the CMC value of the polymer micelle.

### 2.4. Curcumin Chromatographic Analysis

The standard 10.0 mg Cur was weighed accurately and placed in a 10 mL volumetric flask. Methanol (chromatographic grade) was added for dissolution and a constant volume was shaken. 1 mg/mL standard Cur reserve solution was obtained and stored in the dark at 4 °C. A Cur solution with a concentration of 100 μg/mL was obtained by removing 1 mL of the Cur reserve solution and placing it into another 10 mL volumetric flask, then adding methanol to a constant volume. The Cur solution was scanned by an ultraviolet spectrophotometer at full wavelengths in the range of 200–800 nm. Methanol solution was used as the blank control to find the appropriate absorption wavelength for HPLC analysis.

The chromatographic analysis conditions were as follows: instrument: LC-20AD, Shimadzu, Japan, chromatographic column: (5 µm, 4.6 mm × 250 mm; Merck KGaA, Germany), column temperature: 30 °C, mobile phase: acetonitrile-4% acetic acid solution (48:52), flow rate: 1.0 mL/min, injection volume: 20 μL, detection time: 19 min. The Cur standard solution was prepared and injected into the HPLC instrument according to the above chromatographic conditions. The Cur chromatogram was obtained, and the peak area was recorded. Linear regression was performed with Cur concentration as the abscissa and peak area A as the ordinate, and the standard curve was plotted to obtain the regression equation. During detection, the concentration of Cur was calculated according to the regression equation.

### 2.5. In Vitro Release of Cur@ATRA-SS-HA and Cur@HA Nano Micelles

Three buffer solutions (pH 1.2, pH 6.8 and pH 7.4) were used as release media, and the in vitro release experiment was carried out via the dialysis bag method. A total of 1 mg of API and preparation (API dissolved in a very small amount of methanol and preparation dissolved in 1 mL of purified water) were placed in a dialysis bag (MWCO: 3500). After the two ends of the dialysis bag were tightened, they were placed in 100 mL buffer solution and shaken with a shaker. At 10, 20, 30, 40, 60, 90, 120, 180, 240, 360, 480, 600, 720, and 1440 min, 1 mL of buffer solution was removed with a pipette, and 1 mL of buffer solution was added. The samples were diluted with methanol and detected according to the above HPLC method [24].

### 2.6. GSH Sensitivity of Cur@HA and Cur@ATRA-SS-HA

GSH (10 μM, 100 μM, 1 mM, and 10 mM) was added to the micelles. After shaking in a shaker for 12 h, the particle size of Cur@ATRA-SS-HA micelles was monitored by DLS measurement and compared to micelles without GSH, which served as the control. The responses of Cur@HA micelles to 10 mm GSH were also detected and compared with Cur@ATRA-SS-HA.

### 2.7. Cell Culture

Human keloid fibroblasts (HKF) were cultured in DMEM with 10% FBS, and human oesophageal cancer cells (Eca-109) were cultured in 1640 medium with 10% FBS, both of which were adherent cells. Cells were passaged when the cell density reached 80% and used in the next procedures.

### 2.8. Cellular Targeting of Rho@HA and Rho@ATRA-SS-HA

The fluorescent probe rhodamine B was used to replace the drug, and rhodamine B-loaded targeted nanomicelles (RB@ATRA-SS-HA) were prepared following the preparation process. Eca-109 oesophageal carcinoma cells in the logarithmic growth phase were seeded in a 6-well plate (1 × 105 cells/mL, 2 mL), and the culture medium was discarded after cells adhered. A preparation containing the same rhodamine B was added and incubated for 60 min or 120 min, followed by 3 washes in PBS (pH 7.4), fixation in 4% paraformaldehyde (1 mL) for 20 min, and rinsing with PBS (pH 7.4) 3 times. Finally, 0.5 mL DAPI was added to stain cellular nuclei and incubated at 37 °C for 20 min. After washing 3 times with PBS (pH 7.4), a drop (50 μL) of anti-fluorescence quenching mounting solution was placed on the sample to complete the operation. DAPI-labelled nuclei and red light (rhodamine B) were observed with a fluorescent inverted microscope.

### 2.9. In Vitro Biocompatibility and Antitumor Activity of Cur@ATRA-SS-HA

The biocompatibility of ATRA-SS-HA was evaluated by thiazolyl blue tetrazolium bromide (MTT) assay. HKF and Eca-109 cells in the logarithmic growth phase were seeded in a 96-well culture plate at a density of 5 × 10^3^ cells/well and cultured in a 37 °C, 5% CO_2_ incubator for 24 h. After the cells adhered completely, ATRA-SS-HA solutions with different concentrations (1, 2.5, 5, 25, 50, 100 μg/mL) were added, and untreated blank cells were used as the control. Three parallel treatment groups and a blank group were cultured in separate wells. For cytotoxicity experiments, cells were co-incubated with 0, 0.1, 0.2, 0.5, 1, 2, 5, or 10 μg/mL 5-FU, free Cur or Cur@ATRA-SS-HA. After 48 h of incubation, 20 μL MTT solution (5.0 mg/mL) was added to each well. The supernatant was discarded after further incubation for 4 h, and 100 μL DMSO was added to each well and shaken at constant temperature for 20 min. The absorbance at 570 nm was measured by a microplate reader, and the cell viability was calculated using the following formula:
Cell viability% = (A_treat_ − A_blank_)/(A_control_ − A_blank_) × 100%

### 2.10. In Vivo Pharmacokinetic Testing

200 μL rat plasma and 40 µL bis-demethoxycurcumin solution were mixed to prepare a Cur plasma standard solution. The extract was mixed with 600 μL ethyl acetate and vortexed for 1 min, and the ethyl acetate layer was removed after standing. Extraction was performed twice. The resulting extract was centrifuge at 10,000 rpm for 5 min after the second vortex to remove all of the ethyl acetate. The ethyl acetate from the two extractions was combined and dried with N_2_ at 37 °C. Then, 200 μL methanol solution was added, vortexed for 1 min, and centrifuged at 10,000 rpm for 10 min, and the supernatant was injected according to the chromatographic conditions.

## 3. Results and Discussion

### 3.1. Synthesis of ATRA-SS-HA

ATRA (1.631 mmol) was dissolved in DCM, and EDC (1.957 mmol) and HoBt (1.957 mmol) were added and stirred for 2 h for activation. Cystamine was added to another reaction flask to form a suspension. TEA (39.144 mmol) was added under a N2 environment and stirred for 6 h to form a white milk solution. Then, the mixture containing ATRA was slowly added to the reaction flask of cystamine for 24 h. The reaction was monitored with DCM:MeOH = 10:1 and column purified to obtain the intermediate (114 mg). HA (0.02 mmol) was dissolved in a solvent of DMSO:H_2_O = 1:1, and then EDC (0.264 mmol) and sulfo-NHS (0.264 mmol) were dissolved in the solvent and water, respectively, and added to the HA solution. The mixture was stirred at room temperature for 2 h, and the intermediate (0.264 mmol) synthesized in the previous step was added to the reaction solution, stirred under N_2_ for 24 h, dialyzed and spin-dried to obtain the final ATRA-SS-HA product (Figure 1A). The yield of the first step is 16%, and the yield of the second step is 50%. The retinoic acid part in compound 8 is obvious in the NMR spectrum. The unsaturated hydrogen is mainly concentrated in the chemical shift ppm 5–7.5, and the hydrogen on the saturated carbon is mainly concentrated in the chemical shift ppm 1–3. The chemical shifts of the HA part mainly concentrated in ppm 3–5, and the chemical shifts of the amide part formed in ppm 7–9 were more obvious. The hydrogen signal of the photoamine part overlaps between ppm 2–3 (Figure 1B). Compound 8 has A weak νC=CH absorption at ν3097 cm^−1^, where retinic acid is absorbed but hyaluronic acid is not. In addition to the large absorption at 2362 cm^−1^ between 2000 cm^−1^ and 2500 cm^−1^, compound 8 has a small absorption at 2260 cm^−1^ and 2231 cm^−1^. Compared with the HA spectrum (red), there is no absorption here, indicating that the absorption is due to the presence of accumulated double bond νC=C in vitamin As acid. In addition, 2870 cm^−1^ belongs to the vibration contraction zone of νCH3, and the absorption peak at 2850 cm^−1^ belongs to the absorption band of νCH2. Compound 8 has obvious stretching vibration peak νC=O of amide carbonyl group at 1670 cm^−1^. The hydroxyl ν c-OH absorption peak from the hyaluronic acid fraction of compound 8 was found near 1250 cm^−1^. Infrared spectra showed that the synthesized product was the target compound (Figure 2C).

### 3.2. Preparation and Characterization of Cur@ATRA-SS-HA Nanomicelles

Before the preparation of micelles, the critical micelle concentrations (CMC) of the ATRA-SS-HA and HA micelles was calculated and is shown in Figure 2A,B. The CMC of ATRA-SS-HA was 1.673 μg/mL and that of HA was 1.65 μg/mL. Thus, the concentrations of ATRA-SS-HA and HA should be greater than the corresponding CMC value when preparing micelles. Cur-loaded ATRA-SS-HA/HA micelles (Cur@ATRA-SS-HA & Cur@HA) were prepared according to the method described previously. The particle size, polydispersity coefficient (PDI), and potential were measured using a NanoBrook 90 Plus PALS particle sizer. As shown in Table 1, the particle size distribution of Cur@HA was 147.87 ± 2.71 nm, the PDI value was 0.201 ± 0.011, and the zeta potential was −25.91 ± 0.39 mV. The particle size distribution of Cur@ATRA-SS-HA was 149.87 ± 2.34 nm, the PDI value was 0.198 ± 0.012 and the zeta potential was −27.52 ± 0.46 mV. HA and ATRA in ATRA-SS-HA are negatively charged, and higher ratios of HA and ATRA would make Cur@HA and Cur@ATRA-SS-HA appear negatively charged [25,26]. A negatively charged surface can help the Cur@HA and Cur@ATRA-SS-HA avoid binding to negatively charged mucus components and spreading through the mucus layer [27]. Besides, it contained a high negative charge, which could enhance the stability of the Cur@HA and Cur@ATRA-SS-HA and prevent the Cur@HA and Cur@ATRA-SS-HA from precipitating out during a long time of placement [28]. TEM images showed the microstructure of Cur@ATRA-SS-HA and Cur@HA. The particle size was smaller than that measured by the particle size analyser, which may be caused by the shrinkage of the sample after drying during electron microscope detection (Figure 2C). Finally, the stability of the two formulations was investigated, as shown in Table 2. The drug loading, encapsulation efficiency, and particle size were measured after placing the formulations at 4 °C and 25 °C for 0, 15, and 30 days. The results showed that there was no significant difference in the indicators of the two preparations, suggesting that the prepared nanomicelles have good stability.

### 3.3. In Vitro Release of Cur@ATRA-SS-HA and Cur@HA Nanomicelles

The drug release abilities of Cur@ATRA-SS-HA and Cur@HA were investigated by using HCl at pH 1.2 and PBS solution at pH 6.8 and pH 7.4 as release media to simulate the gastric acid environment, slightly acidic tumor microenvironment, and normal physiological environment. First, an HPLC detection system for Cur was established, and the regression equation between the Cur concentration and HPLC peak area was obtained. As shown in Figure 3, at pH 1.2, 6.8, and 7.4, the cumulative release ratios of Cur from both Cur@ATRA-SS-HA and Cur@HA micelles were maintained within a stable range of 58.32–65.98%. Differences in pH sensitivity are not shown. However, due to the poor stability of free Cur in the dissolution medium, only the release behaviour within 720 min was detected, which also showed the potential of nanomicelles to improve the poor stability of free curcumin nanomicelles to improve. In vitro release results showed that the free drug release is faster than from nanomicelles. This might be due to the fact that the API was dissolved in a small amount of methanol, showing a solution state, which can be directly passed through the dialysis bag, while the Cur in the nanomicelles was wrapped in the hydrophobic core, and it took some time to be released. The stronger the interaction between Cur and hydrophobic core, the slower the release rate of Cur [29]. A slow and sustained release of the drug is advantageous because the drug concentration can be maintained for a long time between the lowest effective therapeutic level and the highest tolerable level [30]. 

### 3.4. Cellular Targeting of Cur@ATRA-SS-HA

In a previous experiment, we found that Cur@ATRA-SS-HA and Cur@HA nanomicelles have similar Cur release abilities under different pH conditions. Thus, in the present study GSH was added to the solvent to simulate the tumor microenvironment and to investigate the responsiveness of the nanomicelles. Table 3 shows that the particle size of Cur@ATRA-SS-HA without GSH was 149.11 nm after incubation for 12 h. 10 μM, and 100 μM GSH did not increase the particle size of micelles, whereas 1 mM and 10 mM GSH significantly increased the particle size, resulting in values of 183.43 nm and 263.81 nm, respectively. It is worth noting that 100 μM GSH did not cause an increase in the particle size of Cur@HA, which was similar to Cur@ATRA-SS at 152.73 nm. These data indicate that the GSH-enriched tumor microenvironment breaks disulfide bonds inside micelles, causing micelles to rapidly disintegrate and release the drug housed within them. To further confirm this phenomenon, Cur was replaced with a rhodamine dye and the responsiveness of ATRA-SS-HA to the tumor microenvironment was examined. Rhodamine-loaded non-GSH-responsive HA micelles were used as a control. As shown in Figure 4, the fluorescence intensity of rhodamine in Eca-109 cells of both groups increased with increasing coincubation time, indicating that both formulations released the drug stably during this time period; however, the fluorescence intensity of the ATRA-SS-HA group at the same time was much higher than that of the HA group, confirming that ATRA-SS-HA has good GSH responsiveness and can rapidly release drugs in the tumor microenvironment to achieve rapid targeting to tumor sites.

### 3.5. In Vitro Biocompatibility and Antitumor Activity of Cur@ATRA-SS-HA

The biocompatibility of ATRA-SS-HA was tested using human fibroblast HKFs and oesophageal cancer Eca-109 cells. As shown in Figure 5A,B, for HKF cells, when the concentration of ATRA-SS-HA was 25 μg/mL, the cell viability was still greater than 90%. When the concentration of ATRA-SS-HA reached 50 μg/mL and 100 μg/mL, the survival rate of HKF cells remained between 80–90%. For Eca-109 cells, when the concentration of ATRA-SS-HA was between 0–100 μg/mL, the cell viability remained above 90%, indicating that the formulation had good biocompatibility. The positive control drug, 5-FU, and free Cur were used to detect the cytotoxicity of Cur@ATRA-SS-HA. As shown in Figure 5C,D, the cell inhibition rate of Cur@ATRA-SS-HA at 0.1, 0.5, 0.5, 1, 2, 5, and 10 μg/mL was significantly different from that of free Cur, and the cell effect was better than that of free drug. The IC50 values of the three administration methods are shown in Table 4. The IC50 values of 5-FU, free Cur, and Cur@ATRA-SS-HA were 1.625 μg/mL, 2.588 μg/mL, and 1.136 μg/mL, respectively, reflecting the strong antitumor activity of Cur@ATRA-SS-HA.

### 3.6. In Vivo Pharmacokinetic Testing

First, an HPLC method for the determination of plasma Cur concentration was established. The regression equation between the Cur concentration and peak area is established. The in vivo plasma concentrations of free Cur and Cur@ATRA-SS-HA were investigated. Figure 6 shows that the plasma Cur concentration in the free Cur group gradually increased within 0–1 h and reached the highest value of 0.72 μg/mL at 1 h. The plasma Cur concentration in the Cur@ATRA-SS-HA group reached a maximum of 3 μg/mL at 1.5 h, which was much greater than that of the free Cur group. The plasma concentration of Cur in the free Cur group was <0.1 μg/mL at 12 h, and the Cur concentration in the Cur@ATRA-SS-HA group was >0.1 μg/mL at 48 h. The AUC of the Cur@ATRA-SS-HA group was significantly higher than that of the free Cur group, suggesting that the nanomicelles had excellent pharmacokinetic behaviour and greatly improved the efficacy of curcumin.

## 4. Conclusions

In this study, a GSH-sensitive material, ATRA-SS-HA, was designed and self-assembled with Cur to form Cur@ATRA-SS-HA nanomicelles. The particle size of the micelle was approximately 150–160 nm. The nanomicelles demonstrated excellent drug loading, encapsulation rate, stability, and biocompatibility. The release behaviour was stable at pH 1.2, 6.8 and 7.4. In the tumor microenvironment, GSH induced disulfide bond rupture in Cur@ATRA-SS-HA, promoted the release of Cur, and improved tumor targeting. The IC50 value was significantly lower than that of free Cur and better than that of the positive control drug, 5-FU. In the in vivo pharmacokinetic experiments, the drug-loaded nanomicelles had a higher C_max_, longer half-life, and significantly higher AUC than the free drug group. In conclusion, Cur@ATRA-SS-HA nanomicelles have potential for clinical application.

## Figures and Tables

**Figure 1 pharmaceutics-14-01802-f001:**
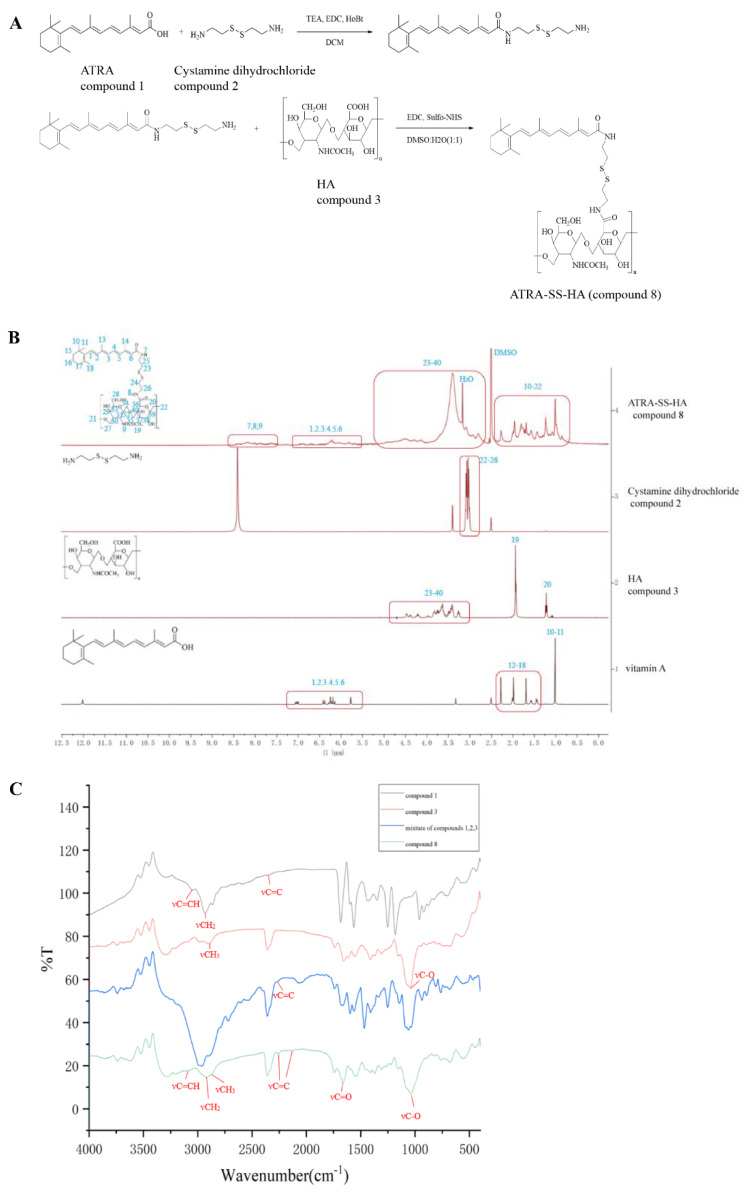
(**A**) Synthesis of ATRA-SS-HA. (**B**) ^1^H-NMR spectrum of ATRA-SS-HA (compound 8), unmodified HA (compound 3), vitamin A, and Cystamine dihydrochloride (compound 2). (**C**) FTIR of ATRA (compound 1), compound 3, mixture of compound 1,2,3 and compound 8.

**Figure 2 pharmaceutics-14-01802-f002:**
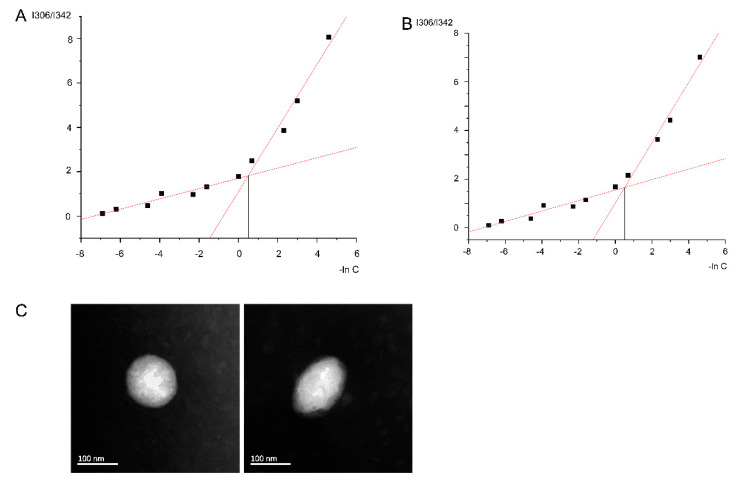
Critical micellar concentration (CMC) determination of HA (**A**) and ATRA-SS-HA (**B**) nano micelles. (**C**) TEM image of Cur@HA(L) and Cur@ATRA-SS-HA(R) nano micelles. Scale bar was 100 nm.

**Figure 3 pharmaceutics-14-01802-f003:**
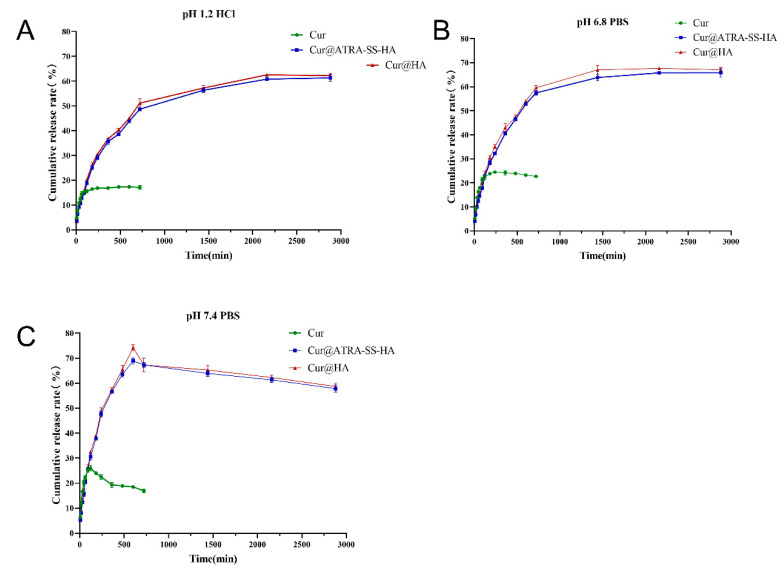
Cumulative release rate of free curcumin, Cur@HA, and Cur@ATRA-SS-HA under pH 1.2 HCl solution (**A**) and pH 6.8&7.4 PBS solution (**B**,**C**).

**Figure 4 pharmaceutics-14-01802-f004:**
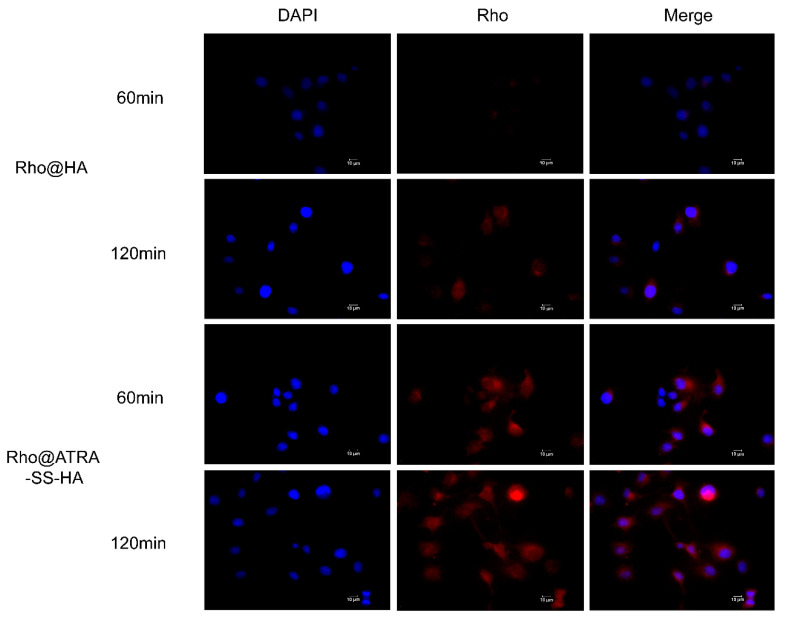
Eca-109 esophageal cancer cells were incubated with Rho@HA and Rho@ATRA-SS-HA for 60 min and 120 min. Images of DAPI and Rhodamine were taken by fluorescence microscope. Scale bar was 10 μm.

**Figure 5 pharmaceutics-14-01802-f005:**
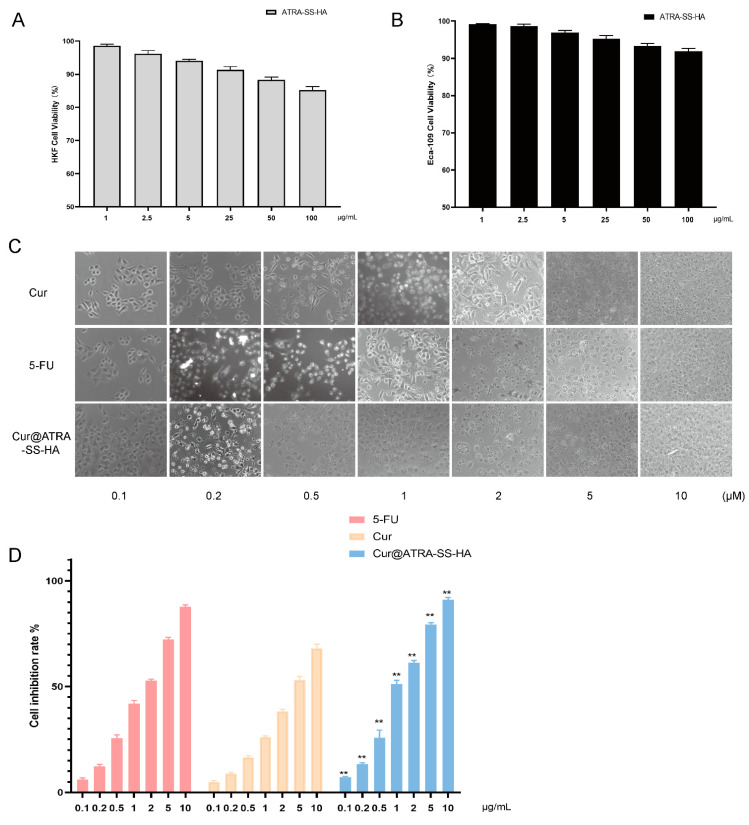
Relative cell viability of HKF cells (**A**) and Eca-109 cells (**B**) after incubation with ATRA-SS-HA at the concentration of 0, 2.5, 5, 10, 50, 100 μg/mL for 24 h. Eca-109 cells were incubated with 5-FU, free curcumin and Cur@ATRA-SS-HA at the concentration of 0.1, 0.2, 0.5, 1, 2, 5, 10 μg/mL for 24 h. Cell image and cell inhibition rate were shown in (**C**,**D**), respectively. **: *p* < 0.01, cell inhibition rate of Cur@ATRA-SS-HA group versus free curcumin group (*n* = 3).

**Figure 6 pharmaceutics-14-01802-f006:**
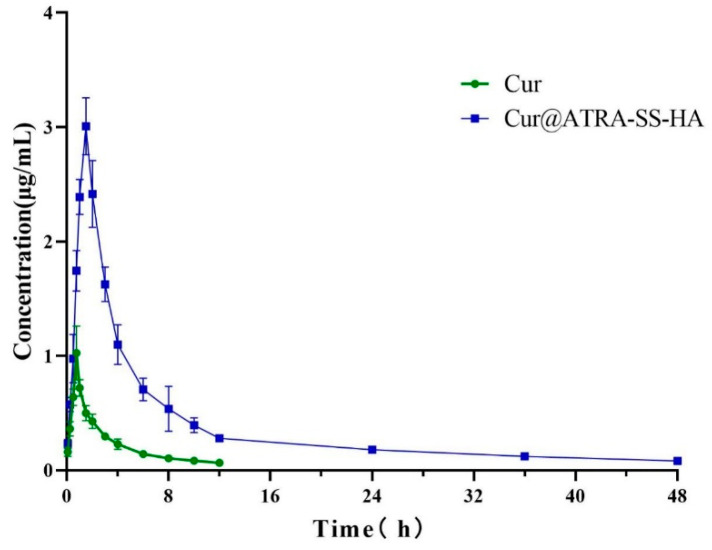
Plasma curcumin concentration at different time points within 48 h after injection of free curcumin and Cur@ATRA-SS-HA.

**Table 1 pharmaceutics-14-01802-t001:** Particle size, PDI and Zeta potential of Cur@ATRA-SS-HA and Cur@ HA.

Sample	Particle Size (nm)	PDI	Zeta Potential (mV)
Cur@ATRA-SS-HA	149.87 ± 2.34	0.198 ± 0.012	−27.52 ± 0.46
Cur@HA	147.87 ± 2.71	0.201 ± 0.011	−25.91 ± 0.39

**Table 2 pharmaceutics-14-01802-t002:** Stability of Cur@ATRA-SS-HA and Cur@ HA.

Sample Time		Drug Loading (%)		Encapsulation Rate (%)		Particle Size (nm)	
Cur@ATRA-SS-HA		4 °C	25 °C	4 °C	25 °C	4 °C	25 °C
	0 day	25.72 ± 1.12	25.59 ± 0.98	90.76 ± 0.46	90.54 ± 0.68	147.26 ± 2.87	147.39 ± 1.78
	15 days	24.12 ± 1.47	23.82 ± 2.11	88.12 ± 1.52	87.10 ± 1.77	150.32 ± 2.98	156.14 ± 2.68
	30 days	23.65 ± 1.86	22.65 ± 2.01	87.23 ± 1.89	85.12 ± 2.53	156.46 ± 3.35	164.58 ± 3.78
Cur@HA		4 °C	25 °C	4 °C	25 °C	4 °C	25 °C
	0 day	25.51 ± 1.82	25.35 ± 0.67	91.16 ± 0.67	91.27 ± 0.56	149.89 ± 2.41	148.17 ± 1.37
	15 days	23.87 ± 1.89	22.19 ± 2.44	88.82 ± 1.64	87.34 ± 1.58	152.17 ± 3.49	155.63 ± 2.56
	30 days	21.92 ± 1.71	21.28 ± 2.58	86.53 ± 1.99	84.48 ± 2.78	157.39 ± 3.88	165.49 ± 3.87

**Table 3 pharmaceutics-14-01802-t003:** Sensitivity of Cur@ATRA-SS-HA and Cur@HA to GSH.

Sample	Particle Size (nm)
10 Μm GSH + Cur@ATRA-SS-HA	146.23 ± 3.18
100 μM GSH + Cur@ATRA-SS-HA	148.62 ± 3.52
1 mM GSH + Cur@ATRA-SS-HA	183.43 ± 4.98
10 mM GSH + Cur@ATRA-SS-HA	263.81 ± 6.84
Cur@ATRA-SS-HA	149.11 ± 2.59
10 mM GSH + Cur@HA	152.73 ± 2.39

**Table 4 pharmaceutics-14-01802-t004:** IC50 value of 5-FU, free curcumin, and Cur@ATRA-SS-HA.

Sample	5-FU	Cur	Cur@ATRA-SS-HA
IC50 (μg/mL)	1.625	2.588	1.136

## Data Availability

Not applicable.

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
