# Peer review of "Design of GSH-Responsive Curcumin Nanomicelles for Oesophageal Cancer Therapy"

_pharmaceutics, 2022, doi:10.3390/pharmaceutics14091802_

Round 1

Reviewer 1 Report (Previous Reviewer 1)

The authors made major revisions and improvements to the manuscript.

Reviewer 2 Report (Previous Reviewer 2)

After the characterization of materials, the manuscript can be accepted. 

This manuscript is a resubmission of an earlier submission. The following is a list of the peer review reports and author responses from that submission.

Round 1

Reviewer 1 Report

This manuscript deals with the design and the development if a GSH-responsive nanomicelles for the delivery of curcumin. This is an interesting inestigation with very good results.

On the other hand, the manuscript need major improvements before publication:

1. Figure 1. The NMR spectra should be removed to the SI section.

2. The physicochemical characteristics of the pure carriers should be presented for comparison reasons. 

3. Explain the negative zeta potential of the prepared systems.

4. Why the authors performed drug release experiments in pH 1.2 and 6.8? Which is the route of administration of this system?

5. Lines 356-377: References should be added.

6. Lines 414-426: References should be added.

The authors should explain their results. Now, they are just describing their data. Please revise the above lines and add references from the literature.

Reviewer 2 Report

In my opinion, the manuscript requires extensive improvement before publishing, therefore, I am suggesting its rejection.

Specific comments:

1) The part about synthesis is a more experimental part, but not a discussion. The spectra didn't provide satisfactory proof to prove the result. The spectra of ATRA should be provided and their physical mixture to prove this claim. I would recommend 2D NMR analysis and FTIR for clarity. What is the yield of this reaction? etc.

2) The stability of the micelles could be checked at 37 oC.

3) Why did the authors check the gastric conditions since the goal was oesophageal cancer?

4) For me, it is weird that free drug release is faster than from nanomicelles. Can you explain?

5) The IC50 values of free drugs and nanomicelles does not differ significantly. Why?